# N-glycosylation in the protease domain of trypsin-like serine proteases mediates calnexin-assisted protein folding

Hao Wang[1,2], Shuo Li[1], Juejin Wang[1†], Shenghan Chen[1‡], Xue-Long Sun[1,2,3,4], Qingyu Wu[1,2,5]*

[1]Molecular Cardiology, Cleveland Clinic, Cleveland, United States; [2]Department of Chemistry, Cleveland State University, Cleveland, United States; [3]Chemical and Biomedical Engineering, Cleveland State University, Cleveland, United States; [4]Center for Gene Regulation of Health and Disease, Cleveland State University, Cleveland, United States; [5]Cyrus Tang Hematology Center, State Key Laboratory of Radiation Medicine and Prevention, Soochow University, Suzhou, China

**Abstract** Trypsin-like serine proteases are essential in physiological processes. Studies have shown that N-glycans are important for serine protease expression and secretion, but the underlying mechanisms are poorly understood. Here, we report a common mechanism of N-glycosylation in the protease domains of corin, enteropeptidase and prothrombin in calnexin-mediated glycoprotein folding and extracellular expression. This mechanism, which is independent of calreticulin and operates in a domain-autonomous manner, involves two steps: direct calnexin binding to target proteins and subsequent calnexin binding to monoglucosylated N-glycans. Elimination of N-glycosylation sites in the protease domains of corin, enteropeptidase and prothrombin inhibits corin and enteropeptidase cell surface expression and prothrombin secretion in transfected HEK293 cells. Similarly, knocking down calnexin expression in cultured cardiomyocytes and hepatocytes reduced corin cell surface expression and prothrombin secretion, respectively. Our results suggest that this may be a general mechanism in the trypsin-like serine proteases with N-glycosylation sites in their protease domains.
DOI: https://doi.org/10.7554/eLife.35672.001

*For correspondence:
wuq@ccf.org

Present address: †Department of Physiology, Nanjing Medical University, Nanjing, China; ‡Human Aging Research Institute, School of Life Sciences, Nanchang University, Nanchang, China

Competing interests: The authors declare that no competing interests exist.

## Introduction

In the human genome, ~2% of the genes encode proteases, among which trypsin-like serine proteases are a major group (*Overall and Blobel, 2007*). Most of the trypsin-like serine proteases act extracellularly to participate in physiological processes, including embryonic development, food digestion, blood coagulation and hormone processing (*López-Otín and Hunter, 2010*; *Neurath, 1984*; *Overall and Blobel, 2007*; *Perona and Craik, 1995*; *Stroud, 1974*). Dysregulated serine protease expression and activity contribute to major health problems such as cardiovascular disease, cancer metastasis, inflammation, and neurological disease (*Craik et al., 2011*; *Gohara and Di Cera, 2011*).

N-glycosylation is a common post-translational modification in proteins (*Eklund and Freeze, 2005*; *Patterson, 2005*; *Varki, 1993*). About two thirds of the predicted human proteins contain N-glycosylation sites (*Apweiler et al., 1999*). Consistently, most of the trypsin superfamily members are N-glycosylated proteins (*Bolt et al., 2007*; *Jiang et al., 2014*; *Liao et al., 2007*; *Miyake et al., 2010*; *Wu and Suttie, 1999*). Many N-glycosylation sites in these serine proteases, especially those in the protease domain, are highly conserved; that is, a specific N-glycosylation site in a protease is conserved not only in the homologs of different species, but also at the same location in other

members of the protease superfamily. Such conservation indicates the functional importance. Indeed, N-glycosylation has been shown to regulate the extracellular expression, secretion and activation of trypsin-like serine proteases, although the underlying mechanisms are not elucidated (*Bolt et al., 2007*; *Gladysheva et al., 2008*; *Jiang et al., 2014*; *Lai et al., 2015*; *Liao et al., 2007*; *Miyake et al., 2010*; *Wu and Suttie, 1999*). It is unclear if N-glycans at the conserved sites have a general role in the biosynthesis of the trypsin-like serine proteases.

Corin is a trypsin-like serine protease that activates natriuretic peptides (*Cui et al., 2012*; *Li et al., 2017*; *Yan et al., 2000*). It consists of a cytoplasmic tail, a transmembrane domain and an extracellular region with multiple protein modules and a C-terminal protease domain (*Hooper et al., 2000*; *Yan et al., 1999*). In cells, corin is made as a zymogen and activated on the cell surface by proprotein convertase subtilisin/kexin-6 (PCSK6) (*Chen et al., 2015*; *2018*). *CORIN* and *PCSK6* variants that impair corin cell surface expression and zymogen activation have been identified in patients with hypertensive diseases (*Chen et al., 2015*; *Cui et al., 2012*; *Dong et al., 2013*; *2014*; *Dries et al., 2005*; *Zhang et al., 2014*; *2017*).

Human corin has 19 N-glycosylation sites in its extracellular region (*Yan et al., 1999*). We and others have shown that N-glycosylation is critical for corin cell surface expression and zymogen activation (*Gladysheva et al., 2008*; *Liao et al., 2007*; *Wang et al., 2015*). Abolishing N-glycosylation sites at Asn80 and Asn231 in the pro-peptide region increased corin shedding on the cell surface, whereas abolishing N-glycosylation site at Asn1022 (N1022), the only N-glycosylation site in the protease domain of human corin, reduced the cell surface expression (*Wang et al., 2015*). To date, how N-glycosylation at N1022 regulates corin cell surface expression remains unknown.

In this study, we made membrane-bound and soluble forms of corin with or without the N1022 N-glycosylation site and analyzed the mutant proteins in transfected cells. We also did proteomic analysis to identify intracellular proteins interacting with corin. We verified our findings in enteropeptidase (also called enterokinase, EK), a transmembrane serine protease, and prothrombin, a secreted serine protease. We found that N-glycosylation in the protease domain of corin, EK and prothrombin has a common role in regulating the extracellular expression of these proteases, which involves calnexin-assisted protein folding and ER exiting.

## Results

### Glycosylation at N1022 promotes cell surface expression of corin zymogen

N1022 is a conserved glycosylation site in the corin protease domain (*Figure 1A*, *Figure 1—figure supplement 1* and *Figure 1—figure supplement 2*). Abolishing this site impairs corin cell surface expression and zymogen activation (*Wang et al., 2015*). To test if the effect is related to zymogen activation, we analyzed corin mutants lacking the activation site (R801A) with or without the N1022 glycosylation site (*Figure 1A*). In western blotting of transfected cell lysates, levels of corin zymogen bands (~160–200 kDa) were similar in corin WT and mutants N1022Q, R801A, and R801A/N1022Q (*Figure 1B*, left). In corin WT, the cleaved protease domain fragment (Corin-p) migrated as an ~40 kDa band under reducing conditions. In the N1022Q mutant, the Corin-p band was lighter and migrated faster, due to the lack of N1022 glycosylation and poor zymogen activation (*Wang et al., 2015*). As expected, no Corin-p band was detected in mutants R801A and R801/N1022Q lacking the activation site. In biotin-labeled cell surface proteins (*Figure 1B*, right), levels of corin bands in the N1022Q mutant were 43 ± 9% of that in WT (p=0.002) and levels in the R801A/N1022Q mutant were 41 ± 8% of that in R801A (p=0.027). The total intensity of WT bands (Corin and Corin-p) was similar to that of R801A (Corin band only). The results indicate that lacking N1022 glycosylation reduces corin cell surface expression with or without the activation cleavage at R801.

### Glycosylation at N1022 promotes soluble corin secretion

The cytoplasmic tail was shown to regulate corin intracellular trafficking (*Li et al., 2015*; *Qi et al., 2011*; *Zhang et al., 2014*). To test if the cytoplasmic and the transmembrane domains are necessary for the N-glycan-mediated corin expression, we tested soluble corin mutants with the Igκ signal peptide with or without mutations at R801 and N1022 (*Figure 1C*). In western blotting of transfected cell lysates, sWT and the mutants sN1022Q, sR801A and sR801A/N1022Q appeared as single bands

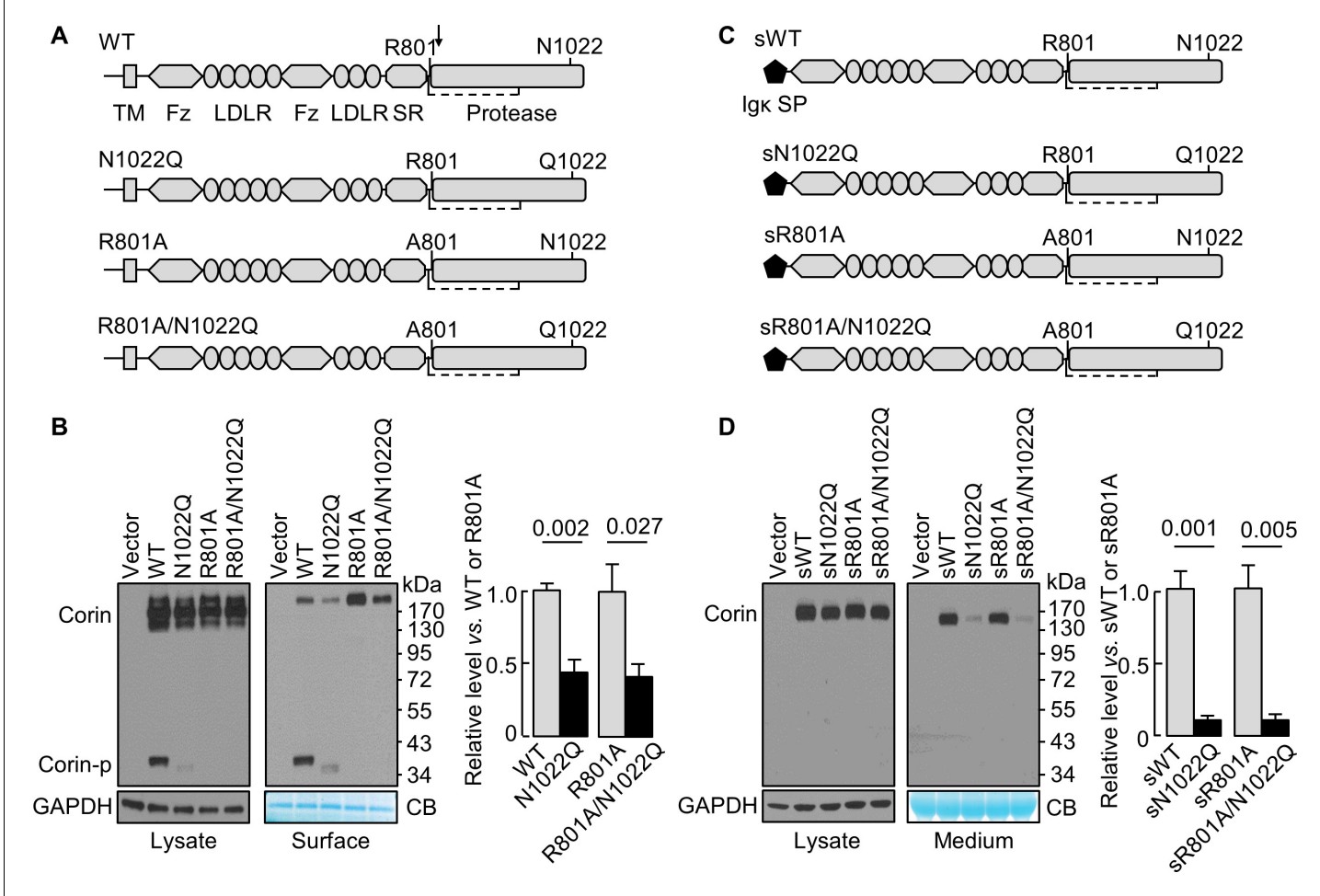

**Figure 1.** N-glycosylation at N1022 in single-chain and soluble corin. (**A**) Illustration of human corin WT and mutants with or without R801 activation site and N1022 N-glycosylation site. TM: transmembrane; Fz: frizzled; LDLR: LDL receptor; SR: scavenger receptor. An arrow indicates the PCSK6-mediated activation cleavage site at Arg801 (R801). A disulfide bond linking the pro-peptide region and the protease domain is indicated by a dashed line. (**B**) Western blotting, under reducing conditions, of corin proteins in lysates (left) or on the cell surface (right) from HEK293 cells. Corin zymogen bands (Corin) and the cleaved protease domain fragment (Corin-p) are indicated. Levels of GAPDH in cell lysates and a Coomassie Blue (CB)-stained non-specific protein in biotin-labeled cell surface proteins were used to assess amounts of proteins in each sample. Relative corin levels on the cell surface are estimated by densitometric analysis of western blots. Data are means ± S.E. from four independent experiments. p-Values are shown in the bar graph. (**C**) Illustration of soluble corin (sWT) and mutants, in which the cytoplasmic and transmembrane domains were replaced by the Igκ signal peptide (SP). (**D**) Western blotting of soluble corin in lysates (left) and medium (right) from HEK293 cells. Levels of GAPDH in cell lysates and a Coomassie Blue (CB)-stained non-specific protein in the conditioned media were used to assess amounts of proteins in each sample. Relative levels of the secreted corin in the medium were estimated by densitometric analysis of Western blots. Data are means ± S.E. from three independent experiments. p-Values are shown in the bar graph.

DOI: https://doi.org/10.7554/eLife.35672.002

The following figure supplements are available for figure 1:

**Figure supplement 1.** The phylogenetic tree of corin proteins in different species.
DOI: https://doi.org/10.7554/eLife.35672.003
**Figure supplement 2.** Alignments of the protease domain of trypsin-like serine proteases.
DOI: https://doi.org/10.7554/eLife.35672.004

at similar levels (*Figure 1D*, left). In the medium (*Figure 1D*, right), levels of sWT and sR801A were similar, whereas levels of sN1022Q and sR801A/N1022Q were 11 ± 2 and 9 ± 4% of sWT and sR801A, respectively, indicating that glycosylation at N1022 promotes soluble corin secretion.

## Glycosylation at N1022 promotes corin exiting from the ER

In Western blotting of lysates from cycloheximide (CHX)-treated cells, levels of WT and the N1022Q mutant decreased over time (*Figure 2A,B*). After 8 hr of CHX treatment, the levels were 7 ± 2% for WT and 32 ± 3% for N1022Q with calculated half-lives of 3.8 ± 0.4 and 6.1 ± 0.3 hr, respectively (p=0.003), indicating that abolishing N1022 glycosylation did not reduce corin protein stability but impaired intracellular trafficking. We digested the proteins with glycosidase Endo H, which removes high-mannose and hybrid N-glycans on proteins in the ER or early Golgi. On western blots, the ratio of Endo H-sensitive *vs.* resistant corin bands was higher in the N1022Q mutant than WT after CHX treatment for 4 hr (*Figure 2C*), indicating that the N1022Q mutant was retained in the ER or early Golgi.

We then co-stained corin and protein disulfide isomerase (PDI) in the cells. Without CHX treatment, WT or N1022Q corin and PDI staining mostly overlapped (*Figure 3A*) with similar Pearson's correlation coefficients (0.49 ± 0.04 and 0.48 ± 0.06, respectively) (*Figure 3B*). After CHX treatment for 4 hr, there was little corin staining in the WT corin-expressing cells, whereas corin staining was strong in the N1022Q-expressing cells (*Figure 3A*, corin (red) vs. PDI (green) ratio in two bottom right panels) with Pearson's correlation coefficients of 0.15 ± 0.06 and 0.35 ± 0.05, respectively (p=0.020) (*Figure 3B*). In co-staining studies for corin and TGN46, a Golgi marker, WT and N1022Q corin had similar distribution patterns with or without CHX treatment (*Figure 3C,D*). These results are consistent with findings from the Endo H experiment, indicating that abolishing N1022 glycosylation prevents corin from exiting the ER.

## Increased N1022Q binding to calnexin and BiP

To identify proteins that interact differentially with corin WT and the N1022Q mutant, we treated the cells with dithiobis succinimidyl propionate (DSP), a protein cross-linker, and did proteomic analysis in samples co-immunoprecipitated with corin. A total of 387 proteins were detected (*Supplementary file 1*). Among the proteins with ≥2 fold differences between WT and N1022Q were calnexin and BiP (binding immunoglobulin protein) (*Supplementary file 2*), two ER proteins in glycoprotein folding and quality control (*Hebert et al., 1995*; *Helenius and Aebi, 2001*). Calnexin and BiP levels were 2.1- and 2.0-fold higher, respectively, in N1022Q-derived samples than those in WT (*Supplementary file 2*). In contrast, the ratio for calreticulin, another ER chaperone in glycoprotein folding (*Hebert et al., 1995*; *Helenius and Aebi, 2001*), was 0.88-fold, whereas the ratios for PDI family members A3 and A4 were 1.24- and 1.67-fold, respectively (*Supplementary file 1*).

To show direct interactions between corin and calnexin or BiP, we immunoprecipitated corin in WT- and N1022Q-expressing cells and analyzed co-precipitated proteins by western blotting. Calnexin and BiP levels from N1022Q-expressing cells were 137 ± 9 and 562 ± 82%, respectively, of those from WT (*Figure 4A–C*). In contrast, levels of calreticulin, HSP70 and HSP90 (two ER chaperones), and PDI were all similar between WT and N1022Q (*Figure 4A,D*). In controls, similar levels between WT and N1022Q were found in V5 pull-down samples and total cell lysates (*Figure 4A*). These results indicate that abolishing N1022 glycosylation increases direct corin binding to calnexin and BiP.

## Effects of glucosidase inhibition on corin binding to calnexin and BiP

In calnexin-assisted glycoprotein folding (*Caramelo and Parodi, 2008*; *Helenius and Aebi, 2001*), triglucosylated oligosaccharides on nascent proteins are trimmed by α-glucosidases I and II to monoglucosylated oligosaccharides, allowing calnexin binding to N-glycans to assist protein folding (*Figure 5A*). Calnexin may bind directly to target proteins via protein-protein interactions, but the functional significance is unclear (*Helenius and Aebi, 2001*; *Ihara et al., 1999*). BiP retains poorly folded proteins in the ER (*Figure 5A*). We treated the cells expressing WT corin and N1022Q with 1-deoxynojirimycin (DNJ), which inhibits glucosidase I and II (*Saunier et al., 1982*) (*Figure 5A*). Without DNJ treatment, calnexin and BiP levels in N1022Q-derived samples were 131 ± 7 and 473 ± 19%, respectively, of WT (*Figure 5B–D*). With DNJ treatment, calnexin and BiP levels increased and became similar between the cells expressing WT and the N1022Q mutant (*Figure 5B–D*), indicating that inhibiting glucosidase activities blocked calnexin binding to N-glycans at N1022 and other N-glycosylation sites on corin and impaired calnexin-assisted folding, resulting in increased direct corin binding to calnexin and BiP.

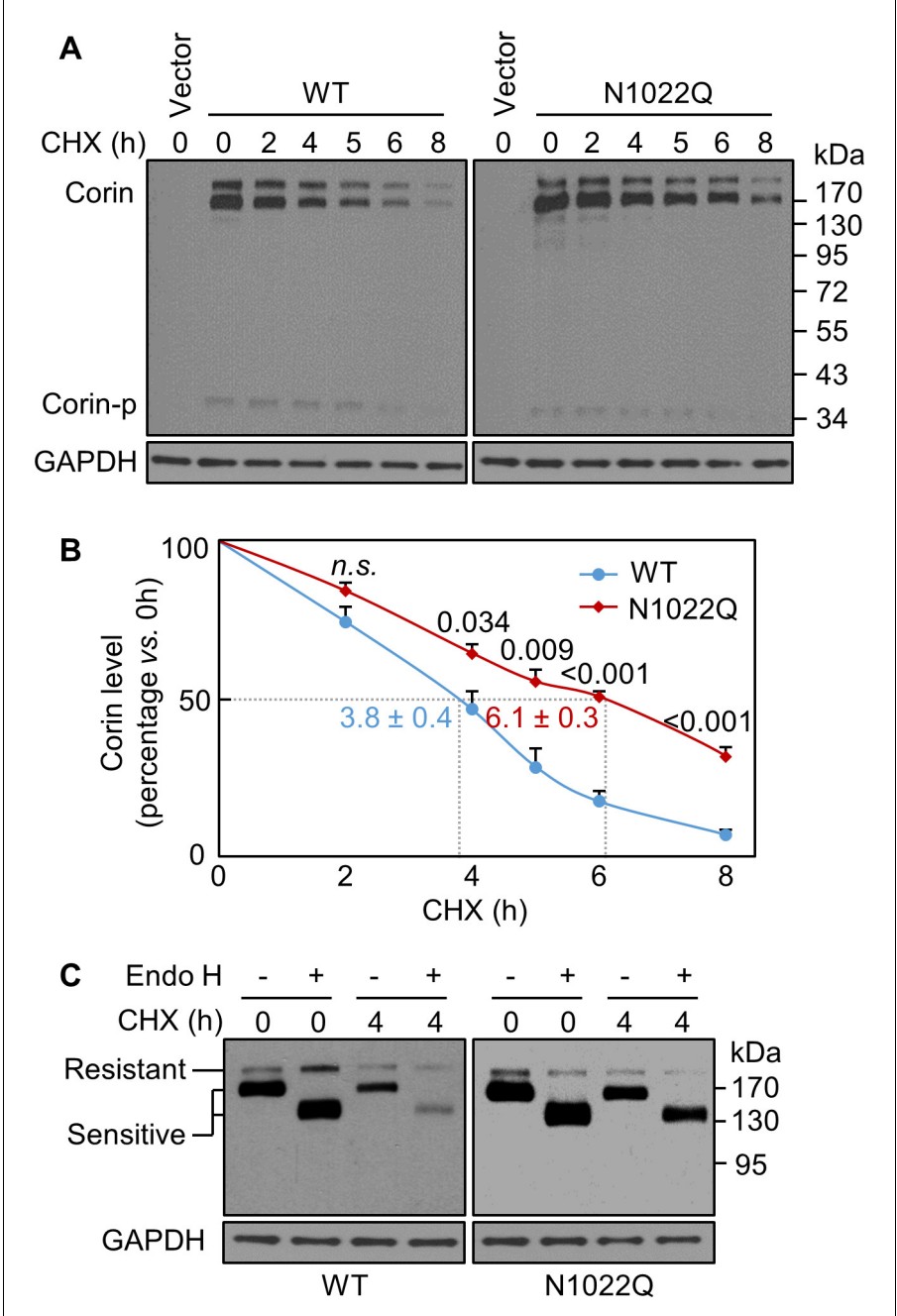

**Figure 2.** Analysis of intracellular corin by CHX-based protein chase and Endo H digestion. (A) Western blotting of corin in HEK293 cells treated without (0) or with CHX over time (h). (B) Percentages of corin WT and the mutant N1022Q levels, with corresponding levels at 0 hr being 100%, were estimated by densitometric analysis of Western blots. In addition to corin zymogen bands (Corin), a weak Corin-p band was detected, which likely represented activated corin on the cell surface. Data are means ± S.E. from four independent experiments. *P* values *vs.* WT at the same time point are shown. n.s.: not significant. The half-lives in h for WT (blue) and N1022Q (red) are indicated. (C) Endo H digestion of proteins from HEK293 cells without (0) or with CHX treatment for 4 hr. Corin proteins without (-) or with (+) Endo H digestion were analyzed by western blotting. Endo H-sensitive and resistant bands are indicated.

DOI: https://doi.org/10.7554/eLife.35672.005

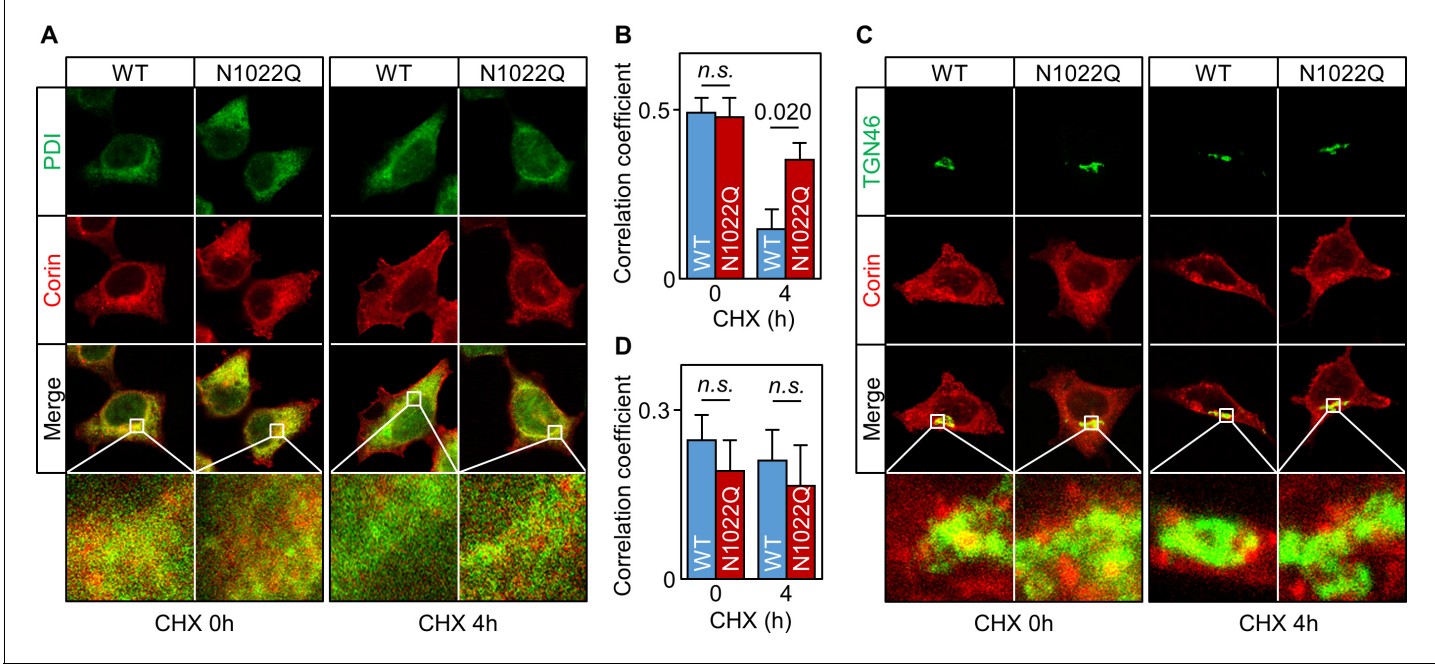

**Figure 3.** Intracellular distribution of corin WT and the N1022Q mutant. (**A**) Co-staining of corin and PDI in HEK293 cells expressing WT corin and the N1022Q mutant without (0) or with CHX treatment for 4 hr. (**B**) Correlation of red (corin) and green (PDI) colors within individual cells was analyzed by Pearson's correlation coefficient. p-Value is shown. n.s.: not significant. (**C**) Co-staining of corin and TGN46 (green) in HEK293 cells expressing WT corin and the N1022 mutant without (0) or with CHX treatment for 4 hr. (**D**) Correlation of red (corin) and TGN46 (green) colors within individual cells was analyzed by Pearson's correlation coefficient. Data are means ± S.E. from five independent experiments.

DOI: https://doi.org/10.7554/eLife.35672.006

## Effect of N-glycosylation on cell surface expression of chimeric proteins

We next made a chimeric protein (CorinEK4N), in which the corin protease domain was replaced by the EK protease domain with four N-glycosylation sites (*Figure 6A*), and additional mutants without the four glycosylation sites (CorinEK4Q) and with (CorinEK4Q/N) a new glycosylation site corresponding to N1022 in corin (*Figure 6A*). On Western blots, CorinEK4N had two major bands (~190 and~220 kDa) (*Figure 6B*). The ~220 kDa band (open arrowhead) was on the cell surface and removable by trypsin before the cells were lysed, whereas the ~190 kDa band (top black arrowhead) was intracellular and resistant to trypsin. In CorinEK4Q, levels of the cell surface protein were 31 ± 5% of CorinEK4N (*Figure 6B*). In CorinEK4Q/N, the level was lower than that in CorinEK4N (49 ± 8%), but higher than that in CorinEK4Q (*Figure 6B*). To exclude the possibility that low levels of the cell surface chimeric proteins were due to increased shedding, we examined the shed proteins in the medium. Levels of CorinEK4Q and CorinEK4Q/N were 8 ± 1 and 29 ± 4%, respectively, of that in CorinEK4N (*Figure 6C*). These results indicate that the function of N-glycans in the protease domain in promoting cell surface expression is not unique to corin.

## N-glycosylation in EK and prothrombin protease domains

We next studied EK (*Kitamoto et al., 1994*), a transmembrane serine protease, and prothrombin (*Wu et al., 1991*), a secreted serine protease. We made EK mutant (EK-4Q) and prothrombin mutant (PT-N416Q) without N-glycosylation sites in the protease domains (*Figure 7A,B*). On western blots (*Figure 7C,D*), EK-4Q and PT-N416Q bands migrated faster than those in EK-WT and PT-WT. Levels of trypsin-removable EK-4Q band on the cell surface, which migrated much closer to the intracellular band due to the loss of 4 N-glycosylation sites, were 14 ± 1% of EK-WT (*Figure 7C*). Levels of PT-WT and PT-N416Q in cell lysates were similar, but the level of PT-N416Q in the medium was 56 ± 30% of PT-WT (*Figure 7D*). These results indicate that N-glycans in the protease domain are important for EK cell surface expression or prothrombin secretion.

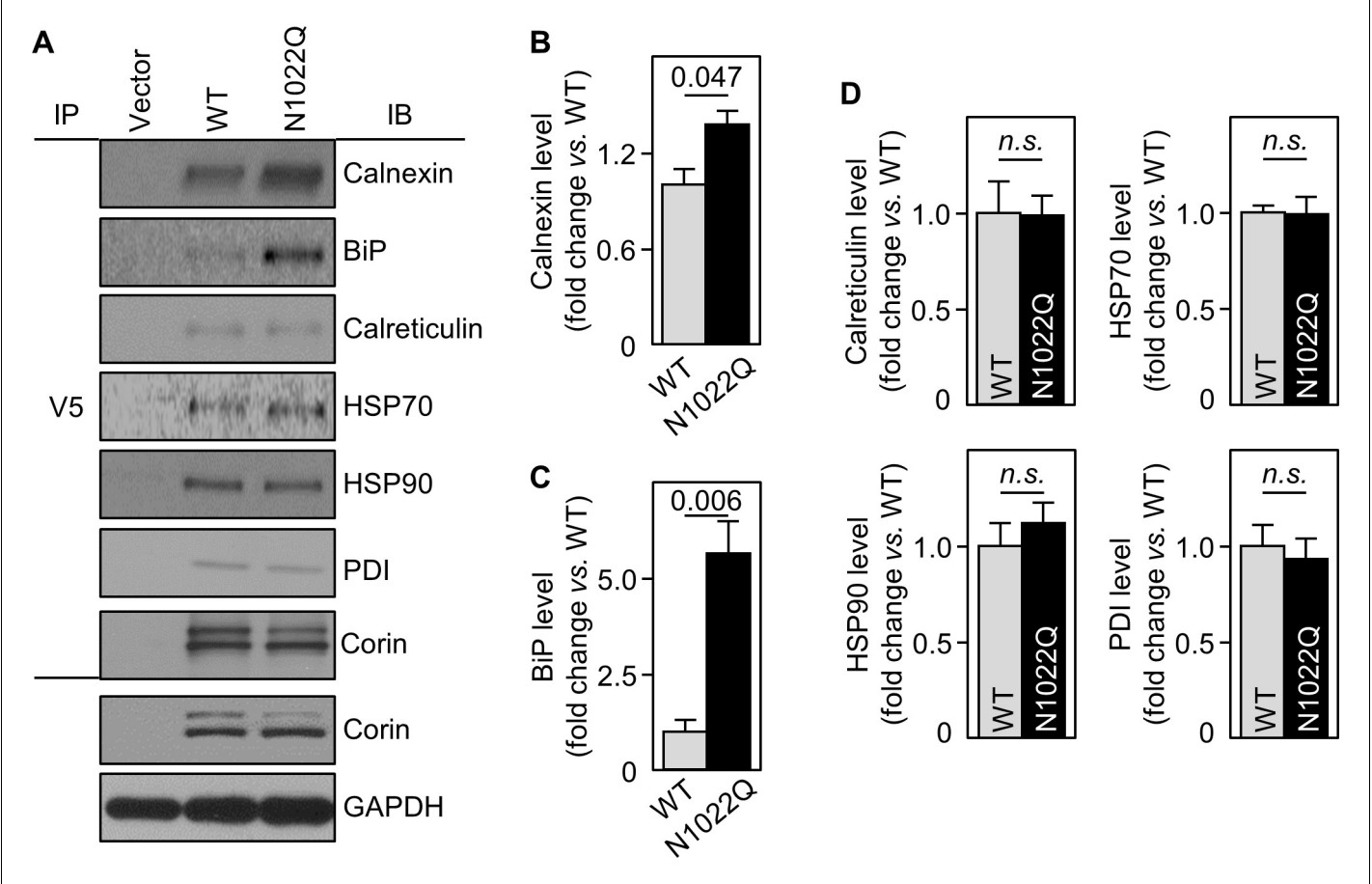

**Figure 4.** Interactions between corin and ER chaperones. (A) Corin proteins in HEK293 cells expressing WT corin and the N1022Q mutant were immunoprecipitated (IP) with an anti-V5 antibody that recognizes the C-terminal V5 tag in corin. Chaperones that co-precipitated with corin were analyzed by immunoblotting (IB, top six panels). Corin in V5 pull-down samples was verified. Corin and GAPDH in the cell lysates were analyzed as additional controls (bottom two panels). Relative levels of calnexin (B), BiP (C), calreticulin, HSP70, HSP90 and PDI (D) were estimated by densitometric analysis of western blots. Data are means ± S.E. from three independent experiments. p-Values are shown in bar graphs. n.s.: not significant.
DOI: https://doi.org/10.7554/eLife.35672.007

## N-glycans in EK and prothrombin protease domains interact with calnexin and BiP

In co-immunoprecipitation and western blotting, calnexin levels in EK-4Q- and PT-N416Q-expressing cells were 165 ± 12 and 171 ± 8%, respectively, of those in respective WT controls (*Figure 8A,B*). BiP levels were also higher in EK-4Q- and PT-N416Q-expressing cells (*Figure 8A,B*). In contrast, cal-reticulin levels were similar in EK-4Q and PT-N416Q compared with corresponding WT controls. In other controls, EK and PT levels in V5 pull-down samples were similar between the WTs and mutants (*Figure 8A,B*). In DNJ inhibition studies (*Figure 8C,D*), calnexin and BiP levels increased in all samples. There were no significant differences in calnexin and BiP levels between the DNJ-treated cells expressing ET-WT and EK-4Q or PT-WT and PT-N416Q. These results indicate a general function of N-glycans in the protease domain in trypsin-like proteases in calnexin-assisted protein folding.

## Effects of DNJ treatment and calnexin knockdown in cardiomyocytes and hepatocytes

We verified our findings in murine HL-1 cardiomyocytes and human HepG2 hepatocytes expressing endogenous corin and prothrombin, respectively. In DNJ-treated HL-1 cells, cell surface corin levels were 26 ± 10% of untreated controls, as estimated by western blotting and densitometry (*Figure 9A*). In DNJ-treated HepG2 cells, prothrombin levels in lysates were similar to untreated

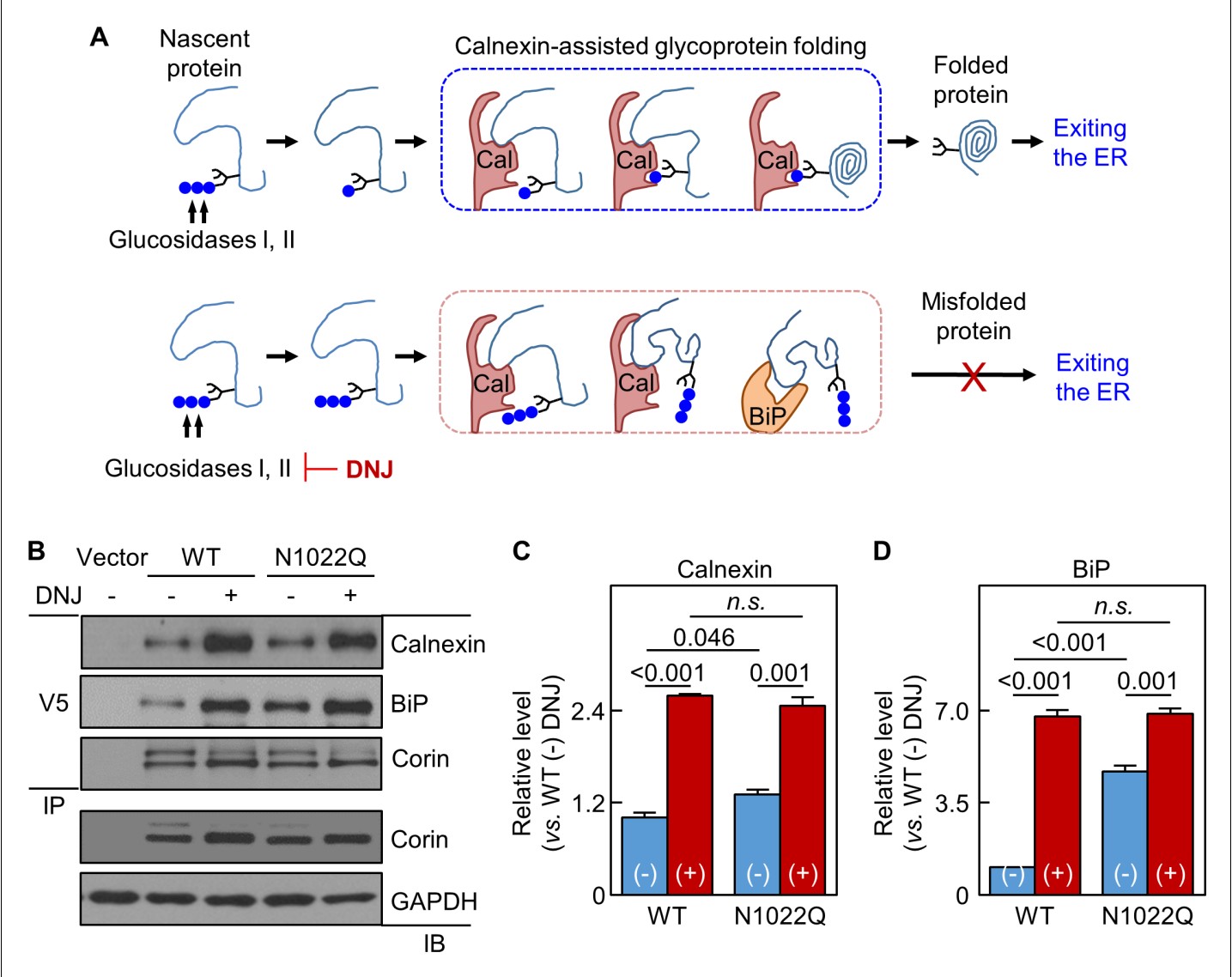

**Figure 5.** Analysis of calnexin interaction. (**A**) A model of calnexin-assisted glycoprotein folding. Cal: calnexin; blue dots: glucose residues. DNJ inhibits glucosidases I and II. (**B**) Co-immunoprecipitation (IP) and western blotting (IB) of corin associated calnexin and BiP in HEK293 cells expressing WT corin or the N1022Q mutant without (-) or with (+) DNJ treatment (top two panels). Corin in V5 pull-down samples was verified (third panel). Corin and GAPDH in the lysates were also verified (bottom two panels). Relative calnexin (**C**) and BiP (**D**) levels were estimated by densitometric analysis of western blots. Data are means ± S.E. from three independent experiments. p-Values are shown in bar graphs. n.s.: not significant.
DOI: https://doi.org/10.7554/eLife.35672.008

controls, whereas the level in the conditioned medium was ~53% of untreated control medium, as measured by ELISA (*Figure 9B*). We next knocked down calnexin expression in HL-1 and HepG2 cells using siRNAs targeting murine and human calnexin genes, respectively. Reduced calnexin protein levels in those cells were verified by western blotting (*Figure 9C,D*). Western blotting and ELISA analyses showed reduced levels of cell surface corin and prothrombin in the conditioned medium, respectively, in HL-1 and HepG2 cells, in which calnexin expression was knocked down (*Figure 9C, D*).

## Discussion

N-glycosylation is important in protein expression and function (*Dalziel et al., 2014*; *Hart and Copeland, 2010*; *Moremen et al., 2012*). Previously, N-glycosylation at N1022 was found to be critical

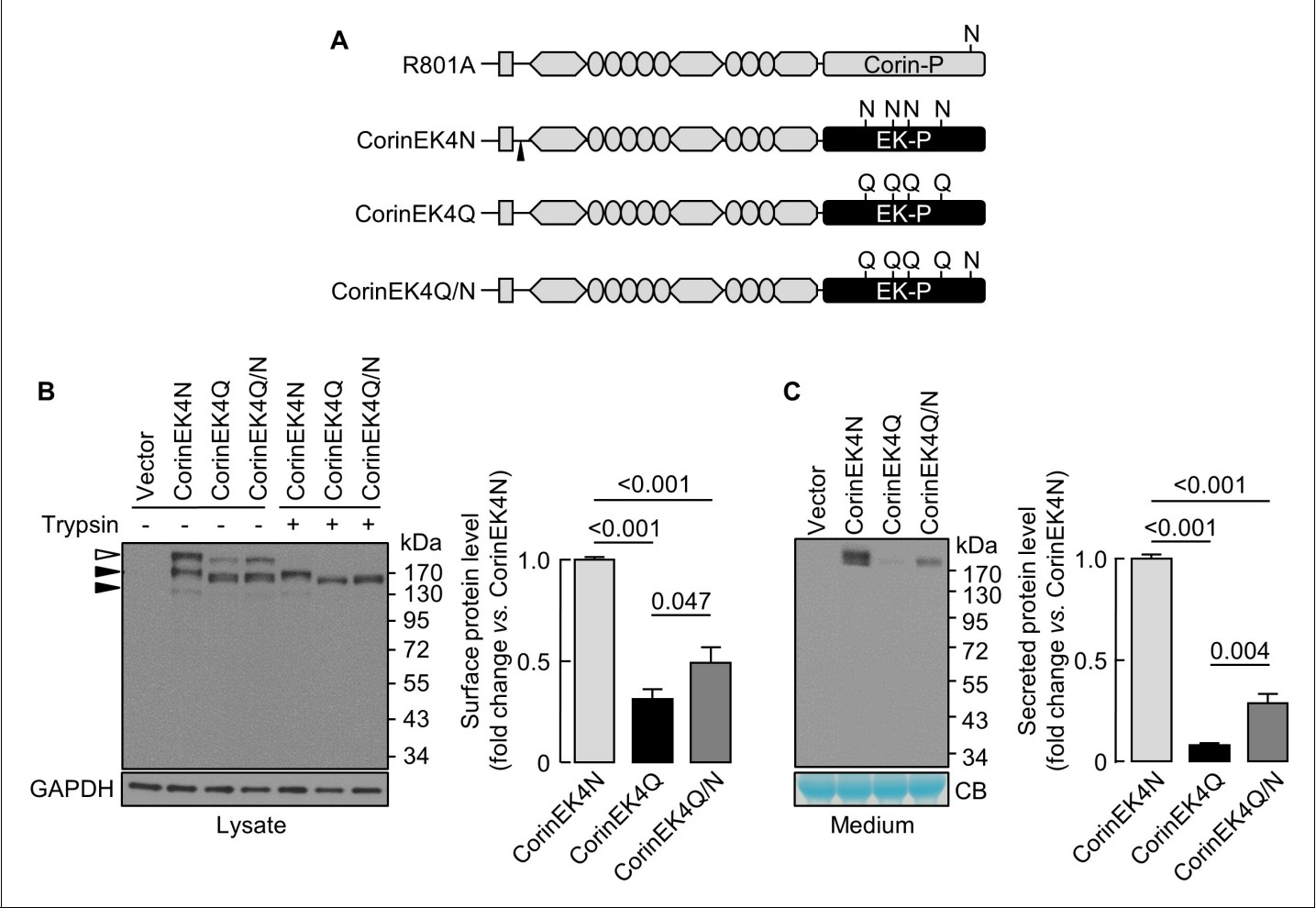

**Figure 6.** Analysis of N-glycosylation in the protease domain of corin-EK chimeras. (**A**) In CorinEK4N, the corin protease domain (Corin-P) was replaced by the EK protease domain (EK-P). The ADAM10-mediated shedding site is indicated by an arrowhead. In CorinEK4Q, all four N-glycosylation sites in the EK protease domain were mutated by Gln (Q) residues. In CorinEK4Q/N, a new N-glycosylation site corresponding to N1022 in corin was added to CorinEK4Q. (**B**) Western blotting of CorinEK4N, CorinEK4Q and CorinEK4Q/N in transfected cells treated without (-) or with (+) trypsin before the cells were lysed. GAPDH levels in cell lysates were used to assess amounts of proteins in each sample. (**C**) Western blotting of shed corin fragments the in medium. Corin levels on the cell surface (**B**) and in the medium (**C**) were estimated by densitometric analysis of western blots. In (**C**), levels of a Coomassie Blue (CB)-stained non-specific protein were used to assess amounts of proteins in each sample. Data are means ± S.E. from at least three independent experiments. p-Values are shown in bar graphs.

DOI: https://doi.org/10.7554/eLife.35672.009

for corin cell surface expression, but the underlying mechanism was unknown (*Wang et al., 2015*). In this study, we found that N-glycosylation at this site was important for corin folding and trafficking in the ER. In proteomic analysis, we identified calnexin and BiP, two ER proteins that bound preferably to the N1022Q mutant.

Calnexin acts in glycoprotein folding (*Caramelo and Parodi, 2008*; *Helenius and Aebi, 2001*). Unlike in heat-shock chaperone-mediated protein folding, which involves direct protein-protein binding, calnexin binds to monoglucosylated oligosaccharides on glycoproteins after triglucosylated N-glycans are trimmed by glucosidases I and II. Calnexin also binds to target proteins via direct hydrophobic interactions (*Brockmeier and Williams, 2006*). Such interactions alone, however, are insufficient for glycoprotein folding. We found increased binding of the N1022Q mutant to calnexin and BiP, indicating that N-glycans at N1022 on corin is important for calnexin-assisted protein folding and ER exiting. The results led to a working model, in which calnexin first binds to nascent corin through direct protein-protein interactions. Subsequent binding of calnexin to monoglucosylated

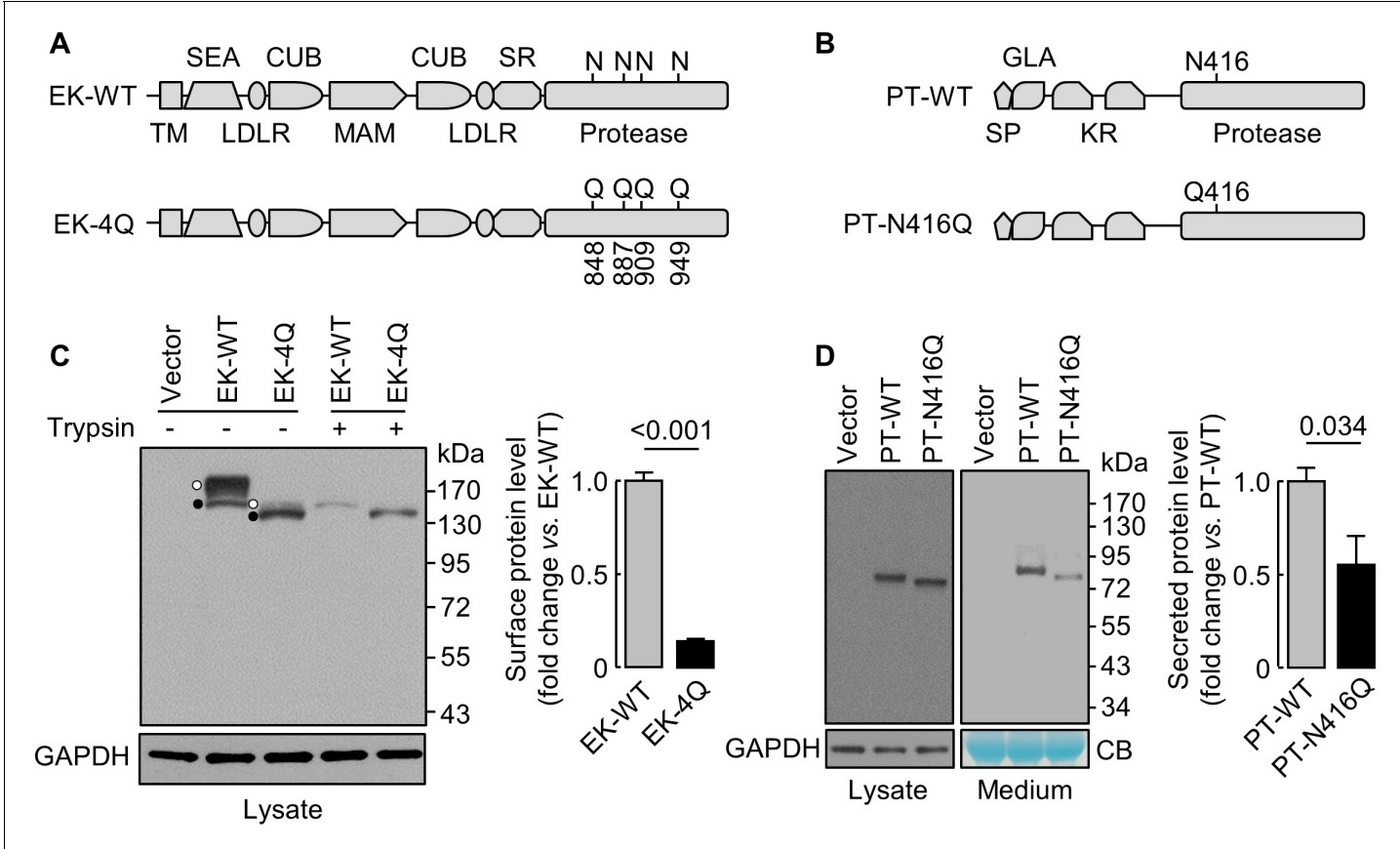

**Figure 7.** Analysis of N-glycosylation sites in the protease domain of EK and prothrombin. (A) Illustration of EK WT and the mutant lacking the indicated N-glycosylation sites (EK-4Q). EK domains include transmembrane (TM), SEA, LDLR, CUB, MAM, scavenger receptor (SR) and protease domains. (B) Illustration of prothrombin WT and the PT-N416Q mutant lacking the N-glycosylation site in the protease domain. Prothrombin domains include signal peptide (SP), Gla (GLA), kringle (KR) and protease domains. (C) Western blotting of EK-WT and EK-4Q in HEK293 cells without (-) or with (+) trypsin treatment before the cells were lysed. The cell surface (trypsin-sensitive; white dots) and intracellular (trypsin-resistant; black dots) bands are indicated. Relative levels of surface EK bands in EK-WT and EK-4Q were estimated by densitometric analysis of western blots. Data are means ± S.E. from four independent experiments. p-Value is shown. (D) Western blotting of PT-WT and PT-N416Q in cell lysates (left) and the medium (right) from HEK293 cells. Relative levels of PT-WT and PT-N416Q in the medium were estimated by densitometric analysis of western blots. Levels of GAPDH in cell lysates and a Coomassie Blue (CB)-stained non-specific protein in the conditioned medium were used to assess protein amounts in each sample. Data are means ± S.E. from four independent experiments. p-Value is shown.
DOI: https://doi.org/10.7554/eLife.35672.010

N-glycans on corin, at N1022 and other N-glycosylation sites, facilitates corin folding. The resultant conformational change in corin disrupts the interaction with calnexin, allowing corin to exit the ER. Consistent with this model, we showed that the treatment of DNJ, a glucosidase inhibitor, increased the binding of the N1022Q mutant and WT corin to calnexin and BiP to similar levels. The results support the importance of N-glycan-calnexin interactions in corin folding and ER exiting. Moreover, the results indicate that N-glycans at other N-glycosylation sites on corin are also involved in the calnexin interaction.

Human corin contains 19 N-glycosylation sites (*Wang et al., 2015*; *Yan et al., 1999*). Among them, N1022 is the only site in the protease domain. Our findings indicate that N-glycosylation in the protease domain is critical for calnexin-assisted folding. In trypsin-like serine proteases, N-glycosylation sites in the protease domain are common. Previously, N-glycosylation in the protease domain of factor VII (FVII) was shown to promote FVII secretion in COS-7 and CHO cells (*Bolt et al., 2007*). Abolishing the N-glycosylation site in the protease domain of chymotrypsin C reduced the secretion in HEK293 cells (*Bence and Sahin-Tóth, 2011*). Conversely, overexpression of a mutant chymotrypsin C lacking the N-glycosylation in the protease domain caused ER stress in cancer cells

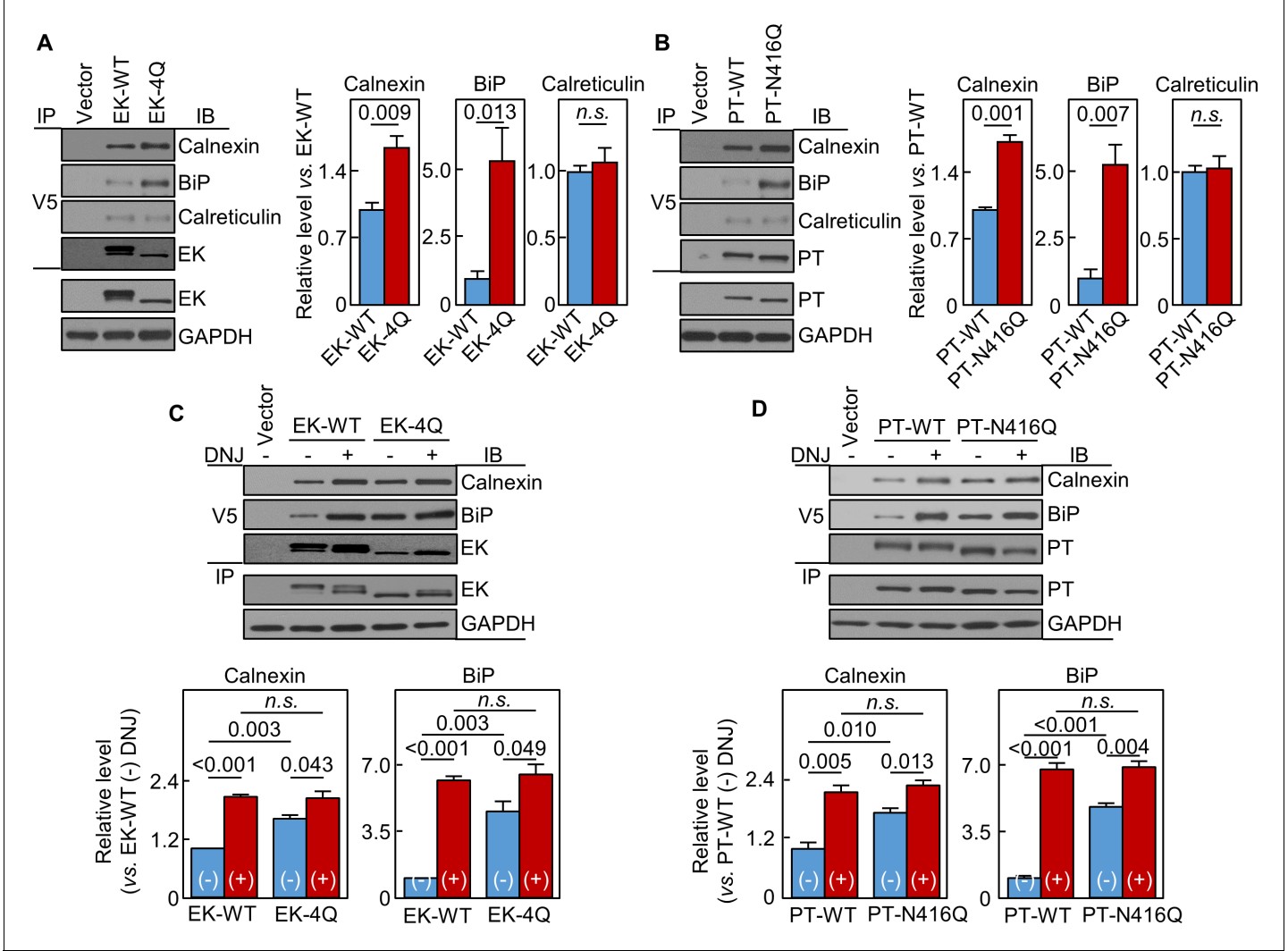

**Figure 8.** Interactions of EK and prothrombin with chaperones. Co-immunoprecipitation (IP) and Western blotting (IB) of EK-WT and EK-4Q (**A**) or PT-WT and PT-N416Q (**B**) binding to calnexin, BiP and calreticulin (top three panels). EK (**A**) and PT (**B**) proteins in V5 pull-down samples were verified. EK, PT and GAPDH in cell lysates were also verified by western blotting (bottom two panels). Relative levels of calnexin, BiP and calreticulin associated with EK-WT and EK-4Q (**A**) or PT-WT and PT-N416Q (**B**) were estimated by densitometric analysis of Western blots. Data are means ± S.E. from three and four independent experiments, respectively. p-Values are shown in bar graphs. IP and IB analysis of EK-WT and EK-4Q (**C**) or PT-WT and PT-N416Q (**D**) binding to calnexin and BiP in the cells without (-) or with (+) DNJ treatment (top two panels). EK (**A**) and PT (**B**) proteins in V5 pull-down samples were verified. EK, PT and GAPDH proteins in cell lysates were also verified. Relative calnexin and BiP levels associated with EK-WT and EK-4Q (**C**) or PT-WT and PT-N416Q (**D**) were estimated by densitometric analysis of Western blots. Data are means ± S.E. from three independent experiments. p-Values are shown in bar graphs.

DOI: https://doi.org/10.7554/eLife.35672.011

(*Bence and Sahin-Tóth, 2011*). These data suggest that N-glycosylation in the protease domain of trypsin-like serine proteases has a general role in calnexin binding and protein folding. Consistent with this hypothesis, DNJ treatment and calnexin knockdown decreased corin cell surface expression and prothrombin secretion in cardiomyocytes and hepatocytes, respectively. Moreover, elimination of N-glycosylation sites in the protease domain of EK or prothrombin increased EK and prothrombin binding to calnexin and BiP and decreased EK cell surface expression or prothrombin secretion in HEK293 cells. These results show that in corin, EK and prothrombin, which have distinct protein domain structures and physiological functions, N-glycosylation in their protease domains has a common function in calnexin-assisted folding and extracellular expression. Possibly, this is a general

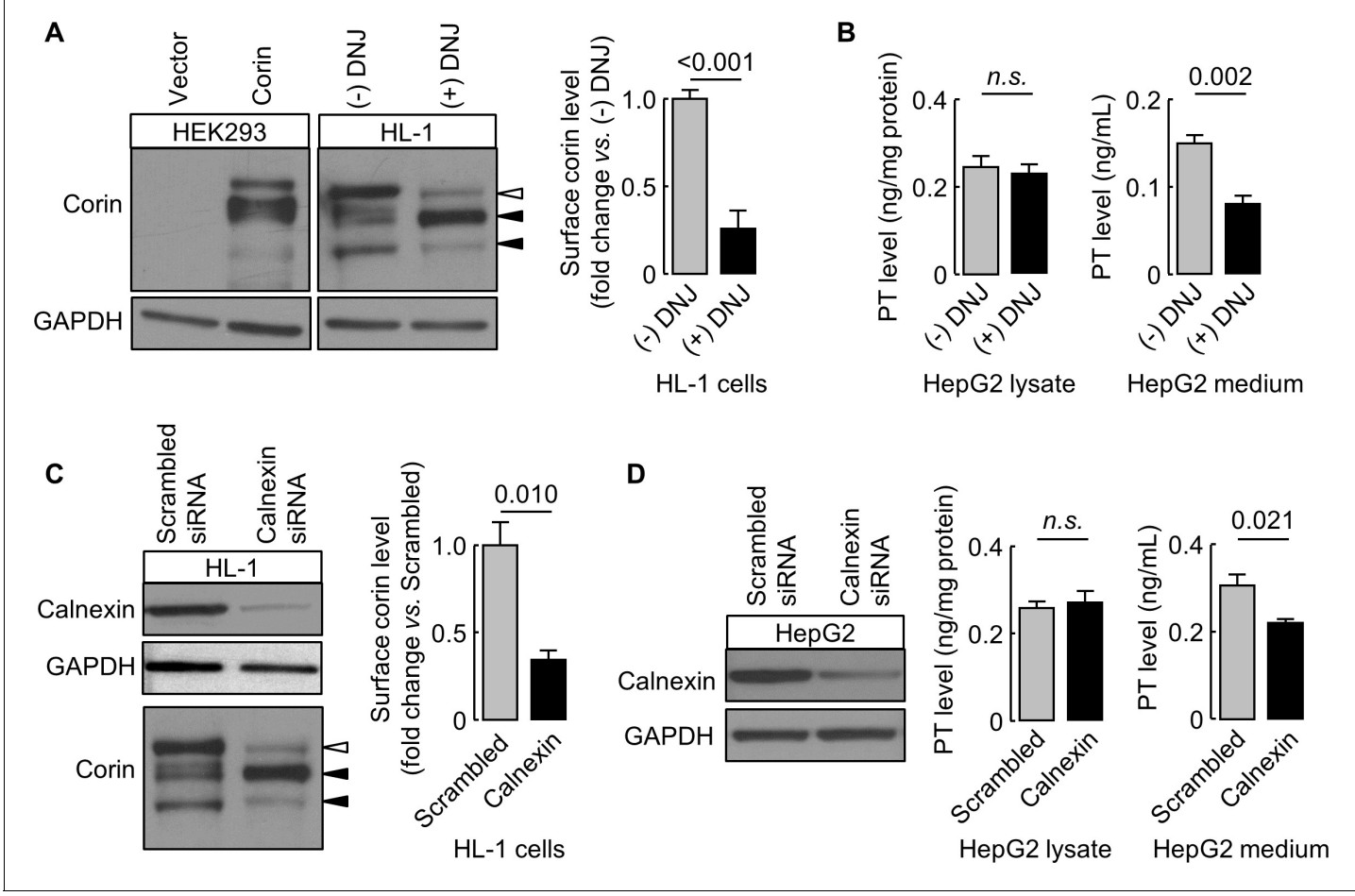

**Figure 9.** Effects of DNJ treatment and calnexin knockdown. HL-1 (**A**) and HepG2 (**B**) cells were cultured without (-) or with (+) DNJ. Recombinant human corin expression in transfected HEK293 cells were included as a control (**A**, left). Corin cell surface expression (**A**) and prothrombin expression in cell lysates and secretion in the medium (**B**) were analyzed by western blotting and ELISA, respectively. Levels of corin cell surface band (open arrowhead) were estimated by densitometric analysis of western blots. Data are means ± S.E. from four independent experiments. p-Values are shown in bar graphs. To knockdown calnexin expression, HL-1 (**C**) and HepG2 (**D**) cells were transfected with calnexin-targeting or control scrambled siRNAs. Calnexin expression levels in the transfected cells were verified by western blotting. Corin cell surface expression (**C**) and prothrombin expression in cell lysates and secretion in the medium (**D**) were analyzed by western blotting and ELISA, respectively. Data are means ± S.E. from three independent experiments. p-Values are shown in bar graphs. n.s.: not significant.

DOI: https://doi.org/10.7554/eLife.35672.012

mechanism in most, if not all, trypsin-like serine proteases that have N-glycosylation sites in their protease domains.

Calreticulin is a soluble calnexin homologous in the ER and acts as a key partner in the calnexin-calreticulin cycle (*Caramelo and Parodi, 2008*; *Ellgaard and Helenius, 2001*; *Helenius and Aebi, 2001*). Like calnexin, calreticulin binds to monoglucosylated oligosaccharides on glycoproteins. In a previous study, inhibition of glucosidase II increased calreticulin binding to cruzipain, a protozoan cysteine protease (*Labriola et al., 1999*). In our study, we found increased binding of N1022Q corin, EK-4Q and PT-N416Q mutants to calnexin but not calreticulin, indicating that calnexin is the primary ER chaperone that interacts with N-glycans in the protease domain of these proteases. If both calnexin and calreticulin recognize similar monoglucosylated N-glycans, how do these proteins distinguish their glycoprotein substrates? Unlike calreticulin, calnexin has a transmembrane domain anchoring calnexin on the ER membrane (*Dalziel et al., 2014*; *Ellgaard and Frickel, 2003*). In most trypsin-like serine proteases, the protease domain is C-terminal. Possibly, the membrane-bound calnexin is more accessible to the N-glycans in the C-terminal protease domain, which comes last from

the translocon on the ER membrane in protein synthesis. This may explain that despite the 18 N-glycosylation sites in the pro-peptide of corin, N-glycosylation at N1022 in the protease domain is required for optimal corin folding and ER exiting. Consistently, N-glycosylation in the protease domain of the CorinEK4N mutant promotes the cell surface expression of the chimeric protein. These results indicate that N-glycans in the protease domain regulate calnexin-assisted folding in a domain-autonomous and calreticulin-independent manner.

The importance of N-glycans in glycoprotein folding varies depending on proteins and cell types (*Helenius and Aebi, 2001*). In the trypsin-like protease superfamily, not all members are N-glycosylated. Some members have N-glycosylation sites in the pro-peptide but not in the protease domain. It is attempting to postulate that N-glycosylation in the protease domain offers an advantage in protein folding efficiency and hence protein production. The requirement of N-glycosylation in a particular protease may depend on its expression level and specific cell environments. More studies are needed to test the folding efficiencies between the trypsin-like proteases with and without N-glycosylation sites in their protease domains in different cells. As trypsin-like proteases are used as biologics to treat human diseases (*Craik et al., 2011*), creation of new N-glycosylation sites may also be a strategy to increase the production of recombinant proteases in vitro.

In summary, we identify a common mechanism of N-glycosylation in the protease domains of corin, EK and prothrombin in calnexin-mediated folding and ER exiting. This process is calreticulin-independent, operates in a domain-autonomous manner, and involves two steps: direct calnexin binding to the target protein and subsequent calnexin binding to monoglucosylated N-glycans. Our findings suggest that this may be a general mechanism in the trypsin-like proteases with N-glycosylation sites in their protease domains. Naturally-occurring mutations disrupting such N-glycosylation sites may impair the expression and function of the trypsin-like serine proteases.

# Materials and methods

**Key resources table**

| Reagent type (species) or resource | Designation | Source or reference | Identifiers | Additional information |
|---|---|---|---|---|
| Gene (*Homo sapiens*) | Corin | NCBI | NM_006587.3 | |
| Gene (*H. sapiens*) | Prothrombin, PT | NCBI | NM_002772.2 | |
| Gene (*H. sapiens*) | Enteropeptidase, EK | NCBI | NM_000506.4 | |
| Genetic reagent (*H. sapiens*) | Calnexin (siRNA kit) | Origene | SR300576 | |
| Genetic reagent (*Mus musculus*) | Calnexin (siRNA kit) | Origene | SR417891 | |
| Cell line (*H. sapiens*) | HEK293 | ATCC | CRL-1573 | STR profiling, no mycoplasma contamination |
| Cell line (*M. musculus*) | HL-1 | PMID: 21518754, EMD Millipore: SCC065 | From Dr. William Claycomb | No mycoplasma contamination |
| Cell line (*H. sapiens*) | HepG2 | ATCC | HB-8065 | STR profiling, no mycoplasma contamination |
| Transfected construct (*H. sapiens*) | Corin plasmid | PMID: 14559895 | | |
| Transfected construct (*H. sapiens*) | sCorin plasmid | This paper | | |
| Transfected construct (*H. sapiens*) | CorinEK plasmid | This paper | | |
| Transfected construct (*H. sapiens*) | EK plasmid | This paper | | |
| Transfected construct (*H. sapiens*) | PT plasmid | This paper | | |
| Antibody | Anti-V5 | Thermo Fisher | R96025 | |
| Antibody | Anti-V5-HRP | Thermo Fisher | R96125 | |

*Continued on next page*

*Continued*

| Reagent type (species) or resource | Designation | Source or reference | Identifiers | Additional information |
|---|---|---|---|---|
| Antibody | Anti-GAPDH | EMD Millipore | MAB374 | |
| Antibody | Anti-PDI | Abcam | ab3672 | Immunostaining |
| Antibody | Anti-TGN46 | Abcam | ab50595 | |
| Antibody | Anti-Igg (mouse)-Alexa-594 | Thermo Fisher | A-21203 | |
| Antibody | Anti-Igg (rabbit)-Alexa-488 | Thermo Fisher | A-11008 | |
| Antibody | Anti-calnexin (human) | Cell Signaling | 2679T | |
| Antibody | Anti-BiP | Cell Signaling | 3177T | |
| Antibody | Anti-calreticulin | Cell Signaling | 12238S | |
| Antibody | Anti-HSP70 | Cell Signaling | 4872T | |
| Antibody | Anti-HSP90 | Cell Signaling | 4877T | |
| Antibody | Anti-PDI | Cell Signaling | 3501T | Western blotting |
| Antibody | Anti-Igg (mouse)-HRP | KPL | 474–1806 | |
| Antibody | Anti-Igg (rabbit)-HRP | KPL | 474–1516 | |
| Antibody | Anti-calnexin (mouse) | Abcam | ab75125 | |
| Antibody | Anti-corin (mouse) | Homemade | PMID: 26259032 | |
| Recombinant DNA reagent | pSecTag/FRT/V5-His Expression kit (vector) | Thermo Fisher | K602501 | |
| Recombinant DNA reagent | pcDNA 3.1/V5-His Expression kit (vector) | Thermo Fisher | K480001 | |
| Commercial assay or kit | ELISA kit (prothrombin) | Abcam | ab108909 | |
| Chemical compound, drug | 1-deoxynojirimycin, DNJ | Alfa Aesar | J62602-MC | |

## Plasmid constructs

The plasmids expressing corin WT and the mutants N1022Q and R801A were described (*Knappe et al., 2003*; *Wang et al., 2015*). Human corin, EK and prothrombin cDNAs were amplified and inserted into pSecTag/FRT/V5-His or pcDNA 3.1/V5-His plasmids (Thermo Fisher) (*Supplementary file 3*) encoding a C-terminal V5 tag. Additional plasmids expressing mutant corin, EK and prothrombin were made by QuikChange II Site-Directed Mutagenesis Kit (Agilent Technologies).

## Cell transfection

HEK293 cells (ATCC, CRL-1573, authenticated by STR DNA profiling, no mycoplasma contamination) were grown in DMEM with 10% fetal bovine serum at 37°C in humidified incubators. At 70–80% of confluency, the cells in six-well plates were transfected with the plasmids using Fugene reagents (Promega). To make stable cells expressing recombinant proteins, the transfected cells were cultured with G418 (400 µg/mL, Teknova). After ~2 w, G418-resistant cells were selected and analyzed by western blotting.

## Western blotting

Recombinant proteins on the cell surface or in the conditioned media and lysates from the transfected cells were immunoprecipitated with an anti-V5 antibody (Thermo Fisher, R96025) and protein A-Sepharose (Thermo Fisher) for western blotting, as described previously (*Wang et al., 2015*). Antibodies used were against V5 (Thermo Fisher, R96125), BiP (Cell Signaling, 3177T), calnexin (Cell Signaling, 2679T), calreticulin (Cell Signaling, 12238S), HSP70 (Cell Signaling, 4872T), HSP90 (Cell Signaling, 4877T) and PDI (Cell Signaling, 3501T). Horseradish peroxidase-labeled secondary antibodies were used (KPL, 474–1806; 474–1516). As a protein loading control for cell lysates, western blots were re-probed with an anti-GAPDH antibody (EMD Millipore, MAB374). As loading controls

for cell surface proteins or proteins from conditioned media, eluted biotin-labeled cell surface proteins or total proteins in the conditioned medium were separated by SDS-PAGE followed with Coomassie Blue staining. Levels of prominent non-specific bands were used to assess similar protein amounts in each sample.

## CHX-based protein chase assay
HEK293 cells expressing corin WT or the N1022Q mutant in six-well plates were incubated with or without CHX (Sigma; 100 µg/mL). The cells were lysed at different time points for western blotting, as described above.

## Endo H digestion
Glycosidase Endo H was used to analyze N-glycans on corin in HEK293 cells. The cell lysates were incubated with Endo H (500 U, New England BioLabs) in 50 mM sodium acetate at 37°C for 1–2 hr. The Endo H-treated proteins were analyzed by Western blotting.

## Immunostaining
HEK293 cells expressing corin were treated with CHX for 4 hr, fixed with 3% paraformaldehyde, permeabilized with 0.2% Triton X-100, and imminostained with antibodies against V5 (1:1000), PDI (1:200, Abcam, ab3672) or TGN46 (1:200, Abcam, ab50595) and Alexa Fluor-594 or 488-labeled secondary antibody (1:1000, Thermo Fisher, A-21203; A-11008). In controls, the primary antibody was replaced by mouse (Thermo Fisher, MA110419) or rabbit (Sigma, I5006) IgG. The stained cells were examined under a confocal microscope (Leica DM2500) and images were analyzed by ImageJ software.

## Protein cross-linking and proteomic analysis
HEK293 cells expressing corin were incubated with dithiobis succinimidyl propionate (DSP) (0.8 mg/mL; Thermo Fisher) at 4°C for 30 min. The reaction was stopped with 0.2 M glycine. Cell lysates were analyzed by immunoprecipitation and SDS-PAGE. Proteins on silver-stained gels were analyzed by liquid chromatography-mass spectrum at the Cleveland Clinic Proteomics Core to identify proteins interacting differentially with corin WT and the N1022Q mutant.

## Glucosidase inhibition
Murine HL-1 cardiomyocytes were a generous gift from Dr. William Claycomb (Louisiana State University Medical Center, New Orleans; no established authentication method for this murine cell line, no mycoplasma contamination), as described previously (*Wang et al., 2008*). Human HepG2 cells were from ATCC (HB-8065, authenticated by STR DNA profiling, no mycoplasma contamination). HL-1, HepG2 and HEK293 cells expressing corin were incubated with 1-deoxynojirimycin (DNJ) (2 mM, Alfa Aesar), which inhibits glucosidases, at 37°C for 24–48 hr. Corin proteins in HL-1 and transfected HEK293 cells were analyzed by western blotting using an antibody against mouse and human endogenous corin (*Chen et al., 2015*). Prothrombin expression in HepG2 cell lysates and the conditioned medium was analyzed by ELISA (Abcam, ab108909).

## Trypsin digestion
To digest cell surface proteins, HEK293 cells expressing corin or EK were incubated with trypsin (0.05%, AMRESCO) at 37°C for 10 min. After washing, cell lysates were prepared for western blotting.

## Effects of calnexin knockdown
To examine effects of calnexin knockdown on corin expression in HL-1 and prothrombin expression in HepG2 cells, siRNAs targeting murine and human calnexin genes (Origene, SR417891 and SR300576) and corresponding scrambled control siRNAs (Origene) were transfected using Lipofectamine reagents (Thermo Fisher). After 24–48 hr, the cells were collected. Calnexin, corin and prothrombin proteins were analyzed, as described above.

## Statistical analysis

The sample size estimation was based on previous studies and pilot experiments. The Student's $t$ test was used to compare two groups with Prism (Graphpad). ANOVA followed by Tukey's post hoc analysis was used to compare three or more groups. A p-value of $< 0.05$ was considered to be statistically significant.

## Acknowledgements

We thank Dr. Belinda Willard for proteomic analysis and Dr. J Evan Sadler (Washington University) for EK plasmid. This work was supported by grants from the NIH (HL126697), the National Science Foundation of China (91639116, 81671485) and Priority Academic Program Development of Jiangsu Higher Education Institutions. The Orbitrap Elite instrument used by the Proteomic Core at the Lerner Research Institute of the Cleveland Clinic was purchased via an NIH shared instrument grant (1S10RR031537-01).

## Additional information

### Funding

| Funder | Grant reference number | Author |
|---|---|---|
| National Institutes of Health | HL126697 | Qingyu Wu |
| The National Science Foundation of China | 91639116 | Qingyu Wu |
| The National Science Foundation of China | 81671485 | Qingyu Wu |

The funders had no role in study design, data collection and interpretation, or the decision to submit the work for publication.

### Author contributions

Hao Wang, Conceptualization, Data curation, Formal analysis, Validation, Investigation, Visualization, Methodology, Writing—original draft, Writing—review and editing; Shuo Li, Juejin Wang, Shenghan Chen, Investigation, Writing—review and editing; Xue-Long Sun, Supervision, Writing—review and editing; Qingyu Wu, Conceptualization, Resources, Data curation, Formal analysis, Supervision, Funding acquisition, Investigation, Visualization, Writing—original draft, Writing—review and editing

### Author ORCIDs

Hao Wang http://orcid.org/0000-0001-6881-9977
Xue-Long Sun http://orcid.org/0000-0001-6483-1709
Qingyu Wu http://orcid.org/0000-0003-0561-9315

### Decision letter and Author response
Decision letter https://doi.org/10.7554/eLife.35672.018
Author response https://doi.org/10.7554/eLife.35672.019

## Additional files

### Supplementary files

• Supplementary file 1. Proteins that differentially bound to WT corin and the N1022Q mutant identified in proteomic analysis.
DOI: https://doi.org/10.7554/eLife.35672.013

• Supplementary file 2. Proteins with a ratio of $\geq 2$ fold between WT corin and the N1022Q mutant.
DOI: https://doi.org/10.7554/eLife.35672.014

• Supplementary file 3. Information of the DNA inserts in the expression plasmids used in this study.
DOI: https://doi.org/10.7554/eLife.35672.015

• Transparent reporting form
DOI: https://doi.org/10.7554/eLife.35672.016

### Data availability

All data generated or analysed during this study are included in the manuscript and supporting files.

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
