## [Decision Letter]

Thank you for choosing to send your work entitled "N-glycosylation in the protease domain of trypsin-like serine proteases mediates calnexin-assisted protein folding" for consideration at *eLife*. Your article has been reviewed by three peer reviewers, and the evaluation has been overseen by a Reviewing Editor, Charles Craik and a Senior Editor, Michael Marletta. The reviewers have discussed the reviews with one another and Charles Craik has drafted this decision to help you prepare a revised submission.

The current work provides new evidence that protease domain N-glycosylation in the trypsin-like serine proteases corin, enteropeptidase and prothrombin is essential for extracellular protease expression and that elimination of protease domain N-glycosylation sites results in endoplasmic reticulum retention through the protein-protein interactions with the chaperone calnexin. The N-glycosylation-mediated folding and trafficking mechanism findings for this important class of enzymes are novel and interesting, experiments are carried out in a thorough, careful manner and the manuscript is well written with helpful, informative schematics that guide the reader through the experimental section.

The manuscript is in principle within the scope of *eLife*'s interest/mission. The reviewers raise valid points and a summary of our assessment follows:

Appropriate controls for the immunoblots and the pull down experiments:

Many of the conclusions in the article are based on subtle differences in protein expression levels as determined by semi-quantitative methods such as immunoblotting and immunocytochemistry. It is therefore important to include loading controls for all immunoblots as well as controls for the amount of protease immunoprecipitated for the V5 pull-downs as outlined by reviewer 1.

Endogenous expression experiments:

The experimental results presented in the manuscript are in heterologous expression systems using recombinantly expressed enzymes. Showing physiological relevance with endogenous protein would increase the biological significance of the findings. As described by reviewer 2, knock-down experiments with siRNA silencing to specifically target calnexin and/or BiP in cell lines that express one or two of the endogenously proteases under study is one approach that could address this concern. A complementary experiment would be DNJ treatment of cell lines expressing endogenous enzymes.

Editorial comments:

The authors are urged to provide greater clarity, particularly for the non-expert on why there is increased binding to N1022Q in the presence of DNJ and why PDI A3 and A4, which also showed enrichment in the N1022Q group proteomics studies were not pursued as pointed out by reviewer 3.

Furthermore, does DNJ increase calnexin (or calreticulin) binding of proteins other than trypsin fold serine proteases or is this is an effect specific to trypsin-fold serine proteases?

Reviewer 1:

This study reports new evidence that protease domain N-glycosylation in the serine proteases corin, EK and prothrombin is essential for extracellular protease expression and that elimination of protease domain N-glycosylation sites results in ER retention through the protein-protein interactions with the chaperone calnexin. The manuscript is well written and points towards new important functions for glycosylation of serine protease domains. Publication of the study is recommended with some corrections and additions.

In general, loading controls for Western blots showing the total amount of protein loaded for each sample in is missing in most of the figures (Figure 1B, D, Figure 2A, C, Figure 6B, C and Figure 7C, D), which makes it difficult to conclude anything with certainty. A Western blot showing the expression level for a housekeeping gene (β-actin, GAPDH or similar) is needed for all blots. Furthermore, when using densiometry tools for quantification, the correct method is normalizing the amount of protein of interest (e.g. corin) to the loading control (e.g. GAPDH), to ensure that the difference in intensity is not due to variation in the amount of total protein loaded.

In Figure 2A and B, the authors conclude that N1022 glycosylation promotes ER trafficking. These data are obtained by quantifying the protein expression levels as determined by the semi-quantitative method Western blotting followed by quantification using densiometric analysis on what appears to be two separate membranes, one for WT and one for N1022Q, which makes it difficult to compare. Although these are commonly used techniques in cell biology it is not a quantitative method and the intensity of the band is not necessarily linear for the protein concentrations detected in Figure 2A. Furthermore, a loading control is needed, as mentioned above. As impaired trafficking of the N1022Q mutant is one of the main conclusions from this study it would be nice to see this observation supported by other more quantitative techniques such as ELISA or similar.

In Figure 2C, the authors conclude that N1022Q mutant is retained in the ER or early Golgi as compared to the WT based on Endo H sensitivity. Again Western blot analysis is used to quantify the amount of protein. The baseline protein expression of the N1022Q mutant appears higher than for the WT (Figure 2C, compare lanes 1 for WT blot with N1022Q blot). Upon deglycosylation at timepoint 0 we see a down-switch in molecular weight for both WT and N1022Q, and the band intensity is comparable for + Endo H and –Endo H. In contrast, at timepoint 4 hours after CHX treatment, there is an inconsistency between the band intensity for WT -Endo H and WT +Endo H? The Endo H treatment should only affect the size of the protein, not the expression level (equivalent levels for N1022Q after 4h CHX – and + Endo H)? Again, a loading control for the blot is needed as mentioned in the first point.

In Figure 3, the authors write that corin N1022Q staining was "strong" compared to "little corin staining" in WT corin expressing cells after 4 h CHX treatment based on ICC analysis. This observation is not evident from the images shown in Figure 4A and B? Looking at Figure 3 there is still (equivalent levels?) corin in the WT (3rd column, 2nd row Figure 3A, red staining) as compared to N1022Q (4th column, 2nd row, Figure 3A, red staining)? It is also difficult to see the difference in overlap with the ER marker PDI and the WT versus the N1022Q as suggested by the authors.

In Figure 4 the authors immunoprecipitate corin by a C-terminal V5 tag and find an increased level of the ER proteins calnexin and BIP in the N1022Q pull down as compared to WT corin pull down. In all previous figures comparing expression levels of N1022Q to WT it looks like there is more N1022Q in the lysate. It would be nice to see the amount of corin precipitated by the V5 tag for both the WT and the N1022Q mutant to rule out that the increased level of calnexin and BiP interaction in the N1022Q pull down is not due to more of the N1022Q protein precipitated as compared to WT.

In Figure 5B a similar issue is found, as described above. It is not evident that the total amount of precipitated corin for the V5 pull down is comparable for WT and N1022Q. A Western blot of V5 immunoprecipitated corin is needed.

In Figure 8 it is similarly important to show the amount of precipitated protease for both EK and prothrombin (PT) in the respective V5 pulldowns to exclude that the difference in calnexin and BiP levels is not due to varying levels of the protease in the samples.

In summary, the study is extensive, systematic and reveals a potential new role for serine protease domain glycosylation in calnexin assisted protein folding and extracellular expression. However, most of the conclusions in the article are based on subtle differences in protein expression levels as determined by semi-quantitative methods such as Western blotting and immunocytochemistry. It is therefore crucial for publication to include loading controls for all Western blots as well as controls for the amount of protease immunoprecipitated for the V5 pull-downs.

Reviewer 2:

In the manuscript by Wang et al., with the title "N-glycosylation in the protease domain of trypsin-like serine proteases mediates calnexinassisted protein folding and extracellular expression" the authors set out to determine how N-glycosylation regulates the extracellular expression, secretion and activation of trypsin-like serine proteases. They report the identification of a common mechanism of N-glycosylation in the protease domains of corin, enteropeptidase and prothrombin in calnexin-mediated glycoprotein folding and extracellular expression. A substantial amount of data is included in the study and experiments are carried out in a thorough manner. The findings are novel and interesting to a wide audience; including scientists interested in basic biology of proteases as well as scientists focusing on post-translational modification of proteins and cellular trafficking.

It is convincingly demonstrated that glycosylation at N1022 promotes corin exiting from the ER. Furthermore, increased N1022Q mutant corin binding to calnexin and BiP is shown. To demonstrate a functional role of calnexin and BiP for corin trafficking, cells were treated with DNJ, a glucosidase inhibitor. This experiment was performed based on the knowledge that in calnexin-assisted glycoprotein folding, triglucosylated oligosaccharides on nascent proteins are trimmed by α-glucosidases I and II to monoglucosylated oligosaccharides, allowing calnexin binding to N-glycans to assist protein folding. DNJ treatment indeed blocked calnexin binding to N-glycans on corin and impaired calnexin-assisted folding, resulting in increased direct corin binding to calnexin and BiP.

1) While the DNJ experiments are informative, it cannot be conclusively established that calnexin and BiP are the essential players in corin glycoprotein folding and transport. The inclusion of selective silencing of calnexin and/or BiP by RNAi would provide valuable data and further substantiate their importance.

2) All experiments are carried out in HEK293 cells using recombinant versions of serine proteases. The biological relevance is therefore unclear. Inclusion of data using cells expression endogenous protease(s) in combination with DNJ treatment and/or RNAi-mediated silencing of calnexin and BiP would greatly enhance the impact of the findings described here.

In conclusion, the manuscript by Wang et al. is thorough and experiments are carried out in a careful manner. The findings are novel and interesting. A knock-down experiment to specifically target calnexin and/or BiP in cell lines that express one or two of the endogenously proteases under study (one or two different cell lines) would significantly enhance the biological relevance of this new exciting N-glycosylation-mediated folding and trafficking mechanism. These experiments would be expected to be manageable, relatively straight forward, and possible to carry out within a reasonable time-frame.

Reviewer 3:

This is well designed and executed study that adds to our understanding of the role of protease domain glycosylation of trypsin fold serine proteases. The results are clean and well interpreted. The manuscript has a good structure and the cartoons in the figures are very helpful to follow the experimental section. There are a couple of questions that need to be properly addressed and are also listed below.

1) Figure 5B shows that DNJ increases calnexin and BiP binding to corin WT and to N1022Q (similar results for EK and prothrombin in later figures). I don't quite understand why there is increased binding to N1022Q in the presence of DNJ, since the N1022-attached glycan (which is absent in the mutant) seems to be the main driver for folding and export. Since the N1022Q mutant has no glycan attached at this position and, therefore, no glucose residues to begin with, it is not clearly understood (by me) why glucosidase I/II inhibition of the N1022Q mutant by DNJ would lead to even more unfolded (and retained) corin in the ER. Since in this case (N1022Q mutant) DNJ inhibits glycosylation at other sites (possibly 18 if I'm correct) the increased calnexin binding seems to be related to impaired trimming of some of these other glycan attachments. How then is the N1022 so important by itself? Maybe I have misunderstood this, but I urge the authors to work on improving the clarity on this section. In addition, the authors may comment in the Discussion as to whether DNJ increases calnexin (or calreticulin) binding of proteins other than trypsin fold serine proteases or whether this is an effect specific to trypsin fold serine proteases (any literature on this?).

2) Based on the proteomics results the authors have focused on calnexin and BiP, even though there are many other hits that came out of this experiment. A potentially relevant binding hit seems to be the PDI A3 and A4, which also showed enrichment in the N1022Q group (Table 1). Any reason why this was not pursued?

---

## [Author Response]

[…] The manuscript is in principle within the scope of eLife's interest/mission. The reviewers raise valid points and a summary of our assessment follows:Appropriate controls for the immunoblots and the pull down experiments:Many of the conclusions in the article are based on subtle differences in protein expression levels as determined by semi-quantitative methods such as immunoblotting and immunocytochemistry. It is therefore important to include loading controls for all immunoblots as well as controls for the amount of protease immunoprecipitated for the V5 pull-downs as outlined by reviewer 1.

Loading controls in all immunoblots and controls for the V5 pull-downs have been included. Please see our responses to reviewer 1.

Endogenous expression experiments:The experimental results presented in the manuscript are in heterologous expression systems using recombinantly expressed enzymes. Showing physiological relevance with endogenous protein would increase the biological significance of the findings. As described by reviewer 2, knock-down experiments with siRNA silencing to specifically target calnexin and/or BiP in cell lines that express one or two of the endogenously proteases under study is one approach that could address this concern. A complementary experiment would be DNJ treatment of cell lines expressing endogenous enzymes.

The DNJ and siRNA silencing experiments have been performed in HL-1 cardiomyocytes and HepG2 hepatocytes, in which endogenous corin and prothrombin are expressed, respectively. The data, which are consistent with the results from HEK293 cells, have been included in new Figure 9. Please see our responses to reviewer 2.

Editorial comments:The authors are urged to provide greater clarity, particularly for the non-expert on why there is increased binding to N1022Q in the presence of DNJ and why PDI A3 and A4, which also showed enrichment in the N1022Q group proteomics studies were not pursued as pointed out by reviewer 3.

Human corin has 19 N-glycosylation sites. In the absence of DNJ, increased binding to calnexin and BiP in the N1022Q mutant, compared to WT corin, indicates the importance of N-glycans at N1022. In the presence of DNJ, all N-glycan-calnexin interactions are expected to be inhibited. Increased binding of N1022Q to calnexin, in the presence of DNJ, indicates that N-glycans at other N-glycosylation sites on corin are also involved in the calnexin interaction. We have revised the Discussion to clarify this point (second paragraph).

In our proteomic studies, we chose 2-fold as the cut-off value for differential binding proteins. We provided this information in the footnote of Supplementary file 2. The N1022Q vs. WT ratios for PDI A3 and A4 were 1.24- and 1.67-fold, respectively, which were below the cut-off value.

However, we do appreciate the Editor and reviewer’s comments. In this revision, we have included the cut-off information in the Results (subsection “Increased N1022Q binding to calnexin and BiP”, first paragraph). We also performed new pull-down experiments using an anti-total PDI antibody (Cell Signaling, 3501T), which showed similar PDI binding between WT corin and N1022Q. The results are included in the revised Figure 4A and D, and the text (see the aforementioned subsection, last).

Furthermore, does DNJ increase calnexin (or calreticulin) binding of proteins other than trypsin fold serine proteases or is this is an effect specific to trypsin-fold serine proteases?

DNJ is a glucosidase inhibitor. As calnexin and calreticulin are involved in the folding process of many glycoproteins, we do not expect the effect of DNJ to be specific to trypsin-like serine proteases. Surprisingly, only limited studies have been published in this area. In our PubMed search, we found only one report, in which DNJ treatment increased calreticulin binding to cruzipain, a protozoa cysteine protease (Labriola et al., 1999). We did not find any papers reporting increased calnexin binding to trypsin-like serine proteases or other proteins in the presence of DNJ. In the revised Discussion, we have cited the Labriola paper (fourth paragraph).

Reviewer 1:[…] In general, loading controls for Western blots showing the total amount of protein loaded for each sample in is missing in most of the figures (Figure 1B, D, Figure 2A, C, Figure 6B, C and Figure 7C, D), which makes it difficult to conclude anything with certainty. A Western blot showing the expression level for a housekeeping gene (β-actin, GAPDH or similar) is needed for all blots.

The GAPDH controls were done in all our original Western blotting experiments. Some of them were not included in the original figures. These controls now have been added to Figure 1B and D, Figure 2A and C, Figure 6B and Figure 7C and D. For Western blots with cell membrane and secreted proteins, we have included Coomassie Blue-stained non-specific protein bands as controls for protein loading (Figure 1B, right, Figure 1D, right, Figure 6C and Figure 7D, right).

Furthermore, when using densiometry tools for quantification, the correct method is normalizing the amount of protein of interest (e.g. corin) to the loading control (e.g. GAPDH), to ensure that the difference in intensity is not due to variation in the amount of total protein loaded.

In our analyses to estimate protein cell surface expression and secretion, levels of recombinant cell surface proteins or secreted proteins were normalized to the total recombinant protein in cell lysates. Loading controls were also verified to ensure proper equal loading.

In Figure 2A and B, the authors conclude that N1022 glycosylation promotes ER trafficking. These data are obtained by quantifying the protein expression levels as determined by the semi-quantitative method Western blotting followed by quantification using densiometric analysis on what appears to be two separate membranes, one for WT and one for N1022Q, which makes it difficult to compare.

In this study, protein levels were normalized to the control sample at 0h from the same experiment and on the same blot. We have revised Figure 2 legend to clarify this.

Although these are commonly used techniques in cell biology it is not a quantitative method and the intensity of the band is not necessarily linear for the protein concentrations detected in Figure 2A. Furthermore, a loading control is needed, as mentioned above. As impaired trafficking of the N1022Q mutant is one of the main conclusions from this study it would be nice to see this observation supported by other more quantitative techniques such as ELISA or similar.

The GAPDH controls have been included in new Figure 2A. The conclusion of impaired trafficking is also supported by immunostaining (Figure 3) and increased calnexin and BiP binding (Figures 4 and 5). We have revised the Results to reflect this point (subsection “Glycosylation at N1022 promotes corin exiting from the ER”, last paragraph).

In Figure 2C, the authors conclude that N1022Q mutant is retained in the ER or early Golgi as compared to the WT based on Endo H sensitivity. Again Western blot analysis is used to quantify the amount of protein. The baseline protein expression of the N1022Q mutant appears higher than for the WT (Figure 2C, compare lanes 1 for WT blot with N1022Q blot). Upon deglycosylation at timepoint 0 we see a down-switch in molecular weight for both WT and N1022Q, and the band intensity is comparable for + Endo H and -Endo H. In contrast, at timepoint 4 hours after CHX treatment, there is an inconsistency between the band intensity for WT -Endo H and WT +Endo H? The Endo H treatment should only affect the size of the protein, not the expression level (equivalent levels for N1022Q after 4h CHX – and + Endo H)? Again, a loading control for the blot is needed as mentioned in the first point.

The GAPDH controls have been added to Figure 2C. Indeed, comparing bands in WT and N1022Q with and without Endo H treatment is less desired. A better way is to compare the ratio of Endo H-sensitive vs. resistant bands within the same lane. Such analysis led to the same conclusion. We have revised the description in the Results (subsection “Glycosylation at N1022 promotes corin exiting from the ER”, first paragraph).

In Figure 3, the authors write that corin N1022Q staining was "strong" compared to "little corin staining" in WT corin expressing cells after 4 h CHX treatment based on ICC analysis. This observation is not evident from the images shown in Figure 4A and B? Looking at Figure 3 there is still (equivalent levels?) corin in the WT (3rd column, 2nd row Figure 3A, red staining) as compared to N1022Q (4th column, 2nd row, Figure 3A, red staining)? It is also difficult to see the difference in overlap with the ER marker PDI and the WT versus the N1022Q as suggested by the authors.

In Figure 3, images in difference columns were from different cells. To ensure the visibility, photo exposure times may not be the same. Thus, images in different columns are not meant to be compared each other. In our analysis, we compared the intensity of corin (red) vs. PDI (green) staining in the same images, i.e. in the same individual cells. This should be clear when one looks at two bottom right panels in Figure 3A. We have revised the Results (subsection “Glycosylation at N1022 promotes corin exiting from the ER”, last paragraph) and Figure 3 legend, to reflect this point.

In Figure 4 the authors immunoprecipitate corin by a C-terminal V5 tag and find an increased level of the ER proteins calnexin and BiP in the N1022Q pull down as compared to WT corin pull down. In all previous figures comparing expression levels of N1022Q to WT it looks like there is more N1022Q in the lysate. It would be nice to see the amount of corin precipitated by the V5 tag for both the WT and the N1022Q mutant to rule out that the increased level of calnexin and BiP interaction in the N1022Q pull down is not due to more of the N1022Q protein precipitated as compared to wt.In Figure 5B a similar issue is found, as described above. It is not evident that the total amount of precipitated corin for the V5 pull down is comparable for WT and N1022Q. A Western blot of V5 immunoprecipitated corin is needed.

We value the reviewer’s suggestion. In revised Figures 4A and 5B, corin proteins in V5-pull down samples are shown.

In Figure 8 it is similarly important to show the amount of precipitated protease for both EK and prothrombin (PT) in the respective V5 pulldowns to exclude that the difference in calnexin and BiP levels is not due to varying levels of the protease in the samples.

EK and PT were verified in the V5 pull-down samples and have been included in revised Figure 8.

Reviewer 2:[…] 1) While the DNJ experiments are informative, it cannot be conclusively established that calnexin and BiP are the essential players in corin glycoprotein folding and transport. The inclusion of selective silencing of calnexin and/or BiP by RNAi would provide valuable data and further substantiate their importance.

We have performed new DNJ and siRNA experiments in HL-1 cardiomyocytes and HepG2 hepatocytes. The results are consistent with the findings from HEK293 cells. The new data are included in new Figure 9.

2) All experiments are carried out in HEK293 cells using recombinant versions of serine proteases. The biological relevance is therefore unclear. Inclusion of data using cells expression endogenous protease(s) in combination with DNJ treatment and/or RNAi-mediated silencing of calnexin and BiP would greatly enhance the impact of the findings described here.

As indicated above, the suggested experiments have been done in HL-1 and HepG2 cells and the results are shown in new Figure 9.

Reviewer 3:…1) Figure 5B shows that DNJ increases calnexin and BiP binding to corin WT and to N1022Q (similar results for EK and prothrombin in later figures). I don't quite understand why there is increased binding to N1022Q in the presence of DNJ, since the N1022-attached glycan (which is absent in the mutant) seems to be the main driver for folding and export. Since the N1022Q mutant has no glycan attached at this position and, therefore, no glucose residues to begin with, it is not clearly understood (by me) why glucosidase I/II inhibition of the N1022Q mutant by DNJ would lead to even more unfolded (and retained) corin in the ER. Since in this case (N1022Q mutant) DNJ inhibits glycosylation at other sites (possibly 18 if I'm correct) the increased calnexin binding seems to be related to impaired trimming of some of these other glycan attachments. How then is the N1022 so important by itself? Maybe I have misunderstood this, but I urge the authors to work on improving the clarity on this section. In addition, the authors may comment in the Discussion as to whether DNJ increases calnexin (or calreticulin) binding of proteins other than trypsin fold serine proteases or whether this is an effect specific to trypsin-fold serine proteases (any literature on this?).

As described above in our replies to the Editorial comments, human corin has 19 N-glycosylation sites. In the absence of DNJ, increased binding to calnexin and BiP in the N1022Q mutant, compared to WT corin, indicates the importance of N-glycans at N1022. In the presence of DNJ, all N-glycan-calnexin interactions are expected to be inhibited. Increased binding of N1022Q to calnexin, in the presence of DNJ, indicates that N-glycans at other N-glycosylation sites on corin are also involved in the calnexin interaction. We have revised the Discussion to clarify this point (second paragraph).

Also described above, in our PubMed search we found only one report, in which DNJ treatment increased calreticulin binding to cruzipain, a protozoa cysteine protease (Labriola et al., 1999). We did not find any papers reporting increased calnexin binding to trypsin-like serine proteases or other proteins in the presence of DNJ. In the revised Discussion, we have cited the Labriola paper (fourth paragraph).

2) Based on the proteomics results the authors have focused on calnexin and BiP, even though there are many other hits that came out of this experiment. A potentially relevant binding hit seems to be the PDI A3 and A4, which also showed enrichment in the N1022Q group (Table 1). Any reason why this was not pursued?

As indicated above in our replies to Editorial comments, we chose 2-fold as the cutoff value for differential binding proteins in our proteomic studies. We provided this info in the footnote of Supplementary file 2. The N1022Q vs. WT ratios for PDI A3 and A4 were 1.24- and 1.67-fold, respectively, which were below the cut-off value. However, we do appreciate the Editor and reviewer’s comments. In this revision, we have included the cut-off information in the Results (subsection “Increased N1022Q binding to calnexin and BiP”, first paragraph). We also performed new pull-down experiments using an anti-total PDI antibody (Cell Signaling, 3501T), which showed similar PDI binding between WT corin and N1022Q. The results are included in the revised Figure 4A and D.